# Kinetics of *Plasmodium* midgut invasion in *Anopheles* mosquitoes

**Gloria Volohonsky**[1]*, **Perrine Paul-Gilloteaux**[2,3,4¤a], **Jitka Štáfková**[1¤b], **Julien Soichot**[1¤c], **Jean Salamero**[2,3,4], **Elena A. Levashina**[1,5]*

**1** INSERM U963, CNRS UPR9022, University of Strasbourg, Strasbourg, France, **2** SERPICO Inria Team/CNRS UMR 144, Institut Curie, Paris, France, **3** National Biology and Health Infrastructure "France Bioimaging", Institut Curie, Paris, France, **4** Cell and Tissue Imaging Facility, IBiSA, Institut Curie, Paris, France, **5** Vector Biology Unit, Max Planck Institute for Infection Biology, Berlin, Germany

¤a Current address: Structure fédérative de recherche santé François-Bonamy, institut de recherche en santé de l'université de Nantes, Nantes, France.
¤b Current address: Institute of Organic Chemistry and Biochemistry, Academy of Sciences of the Czech Republic, Prague, Czech Republic.
¤c Current address: Institute for Parasitology, Zurich University, Zurich, Switzerland.
* gloria.volohonsky@isunet.edu (GV); levashina@mpiib-berlin.mpg.de (EAL)

**Data Availability Statement:** All relevant data are within the manuscript and its Supporting Information files.

## Abstract

Malaria-causing *Plasmodium* parasites traverse the mosquito midgut cells to establish infection at the basal side of the midgut. This dynamic process is a determinant of mosquito vector competence, yet the kinetics of the parasite migration is not well understood. Here we used transgenic mosquitoes of two *Anopheles* species and a *Plasmodium berghei* fluorescence reporter line to track parasite passage through the mosquito tissues at high spatial resolution. We provide new quantitative insight into malaria parasite invasion in African and Indian *Anopheles* species and propose that the mosquito complement-like system contributes to the species-specific dynamics of *Plasmodium* invasion.

## Author summary

The traversal of the mosquito midgut cells is one of the critical stages in the life cycle of malaria parasites. Motile parasite forms, called ookinetes, traverse the midgut epithelium in a dynamic process which is not fully understood.

Here, we harnessed transgenic reporters to track invasion of *Plasmodium* parasites in African and Indian mosquito species. We found important differences in parasite dynamics between the two *Anopheles* species and demonstrated a role of the mosquito complement-like system in regulation of parasite invasion of the midgut cells.

## Introduction

Human malaria is a vector-borne human infectious disease caused by protozoan parasites of *Plasmodium* species. It is widespread in tropical and subtropical regions, including parts of the Americas, Asia, and Africa. Approximately 200 million annual cases of malaria result in half a

**Funding:** EAL received funding by EC FP7 Capacities Specific Programme Research Infrastructures "INFRAVEC" under grant agreement 228421. EAL received funding from the International Associate Laboratory (LIA - Laboratoire International Associé) grant from CNRS. JeS and PPG acknowledge the Structure fédérative de recherche santé François-Bonamy and the SERPICO team, are members of the national infrastructure "France BioImaging" supported by the ANR PIA1 (ANR-10-INBS-04). The funders had no role in study design, data collection and analysis, decision to publish, or preparation of the manuscript.

**Competing interests:** The authors have declared that no competing interests exist.

million deaths [1]. Malaria-causing *Plasmodium* parasites are transmitted by *Anopheles* mosquitoes. Among more than 400 of known *Anopheles* species, only 40 are vectors of human malaria [2].

*Plasmodium* development in the mosquito begins with the ingestion of red blood cells infected with sexual-stage gametocytes. In the mosquito midgut, gametocytes differentiate into gametes that egress from the red blood cells and fuse to form the zygotes that develop into motile ookinetes within 16–18 h. The ookinetes penetrate the midgut epithelium 18–26 h after the infectious blood meal and transform into vegetative oocysts on the basal side of the midgut [3]. After 12–14 days, mature oocysts rupture and release thousands of sporozoites into the mosquito hemocoel. Released sporozoites invade the salivary glands, where they reside inside the salivary ducts to be injected into a new host when the infected mosquito feeds again [4].

The passage of the malaria parasite through the mosquito vector is characterized by a major population bottleneck. Previous studies revealed that mosquitoes kill the majority of invading *Plasmodium* parasites (reviewed by [5,6]), predominantly during the ookinete stage at the basal side of the epithelium [7].

The immune response of mosquitoes to *Plasmodium* parasites is multifaceted and involves multiple processes. In the midgut, reactive oxygen and nitrogen species, hemoglobin degradation products, as well as digestive enzymes and bacterial flora, all affect the rate of *Plasmodium* development (reviewed in [8]). As parasites traverse midgut epithelial cells, the invaded cells produce high levels of nitric oxide synthase and peroxidases, creating a toxic environment for the parasites [9]. As a result, some parasites undergo nitration which marks them for killing by the mosquito complement-like system. Furthermore, intracellular parasites can trigger apoptosis of invaded cells, causing their extrusion and clearance from the cellular layer into the midgut lumen [10]. As *Plasmodium* tries to evade reactive oxygen and nitrogen species inside the cells, these toxic molecules may shape the path taken by the parasite through the cellular layer. The passage of ookinetes through the cellular layer, whether between or through the midgut epithelial cells, was the subject of several studies (reviewed in [3]). These studies concluded that ookinetes always enter into the midgut epithelium intracellularly and that exit from the cellular layer can occur by either an intracellular or intercellular route, depending on extrusion of the invaded cells into the midgut lumen. Indeed, passage through the midgut cells is an obligatory step in parasite invasion as a *P. falciparum* line that could not enter the midgut cellular layer failed to establish infections in *A. albimanus* mosquitoes ([11]).

When the surviving parasites finally reach the basal lamina, they encounter soluble immune factors that circulate in the hemolymph. The complement-like protein TEP1 and leucine-rich repeat proteins APL1C and LRIM1 form a complex that mediates parasite killing [12,13]. Histological studies have shown that parasites crossing the cellular layer can be found both inside and in between midgut cells [3,14]. However, it is not yet known whether some parasites cross the cellular layer exclusively between cells, thus avoiding intracellular nitration and subsequent recognition by TEP1.

Despite accumulating evidence of molecular processes that govern the passage of motile ookinetes through mosquito tissues, the complexity and diversity of this dynamic process remains to be deciphered. Three modes of motility were reported for the invading ookinetes, namely spiraling, gliding and stationary rotation [15,16]. Spiraling and gliding movements result in active displacement of the parasite in space. In contrast, stationary rotation movement was observed for prolonged periods of time and resulted in no displacement of the ookinete. Because of the lack of markers of the entire midgut cellular layer, previous studies did not establish how distinct types of movements correlate with ookinete location in the midgut.

It has been previously demonstrated that *Anopheles* species differ in their vector competence [17]. In the laboratory, *A. stephensi* (*As*) and *A. gambiae* (*Ag*) can be infected with the

murine parasite *P. berghei* (*Pb*) with *As* yielding higher parasite loads than *Ag* [18]. We set out to quantify by imaging *in vivo* migration of the RFP-expressing *Pb* ookinetes through the epithelial cells in these two genetically modified mosquito species that express GFP in the midgut cells. Using high-speed spinning disk microscopy and automated image analyses, we quantified parasite invasion dynamics at high spatial and temporal resolution. Our results uncovered differences in *Pb* invasion of closely-related mosquito species, pointing to important species-specific mechanisms that regulate mosquito–parasite interactions. Moreover, silencing of the major component of the mosquito complement-like system affected the parasite invasion dynamics, suggesting that TEP1 also regulates the early stages of the midgut invasion process.

## Results and discussion

### Effect of TEP1 on midgut invasion of *P. berghei* ookinetes

To study the passage of *Pb* ookinetes through the mosquito midgut, we combined multiscale imaging techniques with high-throughput data analysis and mining (Fig 1). We used transgenic mosquitoes expressing GFP under the mosquito midgut-specific *G12* and *Drosophila Actin5c* promoters [19,20] to label mosquito midgut cells, and transgenic rodent *Pb* parasites expressing RFP under a constitutive promoter [21] (S1A Fig). We first made sure that expression of the reporters did not interfere with *Plasmodium* infection. As expected, a significant difference was observed in infection intensity between *As* and *Ag*. Regardless of the infection levels, *As* developed significantly higher oocysts numbers than *Ag* (S1B Fig). We concluded that the transgenic mosquito and *Pb* lines can be used for *in vivo* imaging.

The transgenic mosquito lines expressed GFP in the entire midgut cell, therefore, we measured the exact position of RFP-expressing parasites relative to the cellular layer (Fig 2). To this end, we collected large series of z-stack images of live parasites inside the dissected mosquito midguts at different time points after infection (S2 Fig) and time-lapse images of selected parasites (S1 and S2 Tables). These tools enabled us to study the parasite invasion process at two time-scales: one was based on statistical analysis of parasites in three dimensional (3D) snapshots of the state of infection between 18 and 25 h post infection (hpi), the second tracked single parasites 18 to 25 hpi over a time of 20 min to 2 h. For each record, marked with a Global Unique IDentifier (guid), parasites and nuclei of the midgut cells were segmented and their positions in 3D space were calculated relative to the cellular layer at each examined time point after infection (Fig 2, S3 Fig). The position of parasites relative to the cellular layer was determined by fitting the midgut cell nuclei position by a cubic spline surface. This surface was then considered as the central position of the cellular layer (normalized z = 0). An average thickness of 5 μm above and below this surface defined the average cellular layer position.

We next examined whether the dynamics of parasite invasion was similar in two *Anopheles* species and measured the number of parasites at each position (blood meal, cellular layer, and basal lamina) in *As* and *Ag*. Analyses of all time points did not detect significant differences in parasite localization between the two species (Fig 2E). The majority of ookinetes were detected in the blood meal (70%) and in the cellular level (20%). Only few ookinetes crossed the midgut and reached the basal side (10%). Interestingly, silencing of the major antiparasitic factor *TEP1* in *Ag* (*AgTEP1*$^{KD}$) significantly ($p$ = 0.001, Mann-Whitney t-test) changed spatial distribution of the parasites with only 40% of ookinetes observed in the blood meal, 45% in the cellular layer and 15% at the basal side. The observed changes in the dynamics of *Pb* invasion in *TEP1*-depleted mosquitoes suggested that in addition to the role of TEP1 in ookinete killing at the basal side, this factor also inhibits earlier stages of ookinete midgut invasion. Previous studies reported *TEP1* expression in the larval gastric caeca and adult midguts [22,23]. In line with these reports, silencing of *TEP1* also affected midgut microbiota by an as yet unknown

**Fig 1. Workflow and experimental settings.** *A. stephensi* (*As*) and *A. gambiae* (*Ag*) mosquitoes were blood-fed on *P. berghei* infected mice, their midguts dissected and visualized using fast confocal microscopy. Images from all experiments collected at different time points after infection were uploaded into an image database and annotated. Quantitative data was extracted from the images in the database regarding the number, position, and intensity of visualized parasites. The results of the data analysis reveal the kinetics of parasite invasion.

mechanism [24]. Furthermore, depletion of APL1 in *As* resulted in altered midgut micro-biome, a change that could affect parasite invasion [25]. Our findings extend these observations to the early stages of parasite invasion and suggest that in addition to parasite killing at the basal side, TEP1 directly or indirectly inhibits *Plasmodium* midgut traversal.

## Dynamics of ookinete midgut invasion

We next focused on *P. berghei* ookinete passage through the mosquito midgut cells at different time points after infection and examined the proportion of parasites at each position (blood meal, cellular layer, and basal lamina). To this end, we calculated the average proportion of parasites at each position at the early (18–20 hpi), intermediate (21–23 hpi) and late (24–25 hpi) intervals after infection (Fig 3A, S4 Fig).

We observed that in *As* mosquitoes the proportion of blood bolus-residing parasites did not change over time. The proportion of parasites within the cellular layer significantly increased from 14 to 32% during the transition between the early (18–20 hpi) and intermediate (21–23 hpi) time intervals. However, this increase did not cause accumulation of the ookinetes at the basal lamina. Instead, a significant decrease from 20 to 4% was detected in the proportion of basally located parasites between the early (18–20 hpi) and intermediate (21–23 hpi) time intervals. We were surprised to see that this decrease was temporal, as the proportion of parasites in the basal lamina significantly increased to 10% at the late time interval (24–25 hpi). A similar decrease in the proportion of basally located ookinetes was detected in *Ag*, where the proportion of parasites at the basal lamina declined from 12% at 18–20 hpi to 3% at 21–23 hpi, and then increased again to 14% at the late time interval.

Since the mosquito immune system targets the ookinetes at the basal side of the midgut [26], we examined whether the observed decrease in the proportion of basally located ookinetes was rescued by *TEP1* knockdown. *TEP1* silencing eliminated the decrease in the basally located ookinetes observed in *Ag* mosquitoes and at the same time increased the proportion of parasites within the cellular layer (Fig 3A). These results suggest that the first invading ookinetes are rapidly killed and lysed by the mosquito immune system. The most parsimonious explanation of the observed parasite accumulation at the basal lamina at later time points may be an asynchronous midgut invasion by *Pb*, where the first wave of invading ookinetes exhausts limited components of the mosquito immune system and, thereby, benefits the establishment of infection by the second wave of the parasites. This hypothesis is in line with the observation that not all parasites are recognized and killed by TEP1 at the basal lamina. We propose that early crossing parasites may serve as pioneers that locally deplete TEP1, allowing later-coming parasites to survive the immune attack, however, further experiments are necessary to validate this hypothesis.

To better understand *Pb* invasion dynamics, we measured ookinete motility in time-lapse experiments. The blood-filled midguts were dissected from infected mosquitoes and mounted *ex vivo* for imaging by spinning disk microscopy for a period of 20 to 120 min. In line with the previous work [15], we observed four distinct ookinete motility modes: (i) passive floating within the blood bolus (guid 2107, guid 1615, S1 Table), (ii) gliding within the cellular layer (guid 1628, S1 Table) (iii) spiraling in the blood meal and within the cellular layer (guid 1622, guid 1624, S2 Table) and (iv) stationary rotation without translocation within the cellular layer

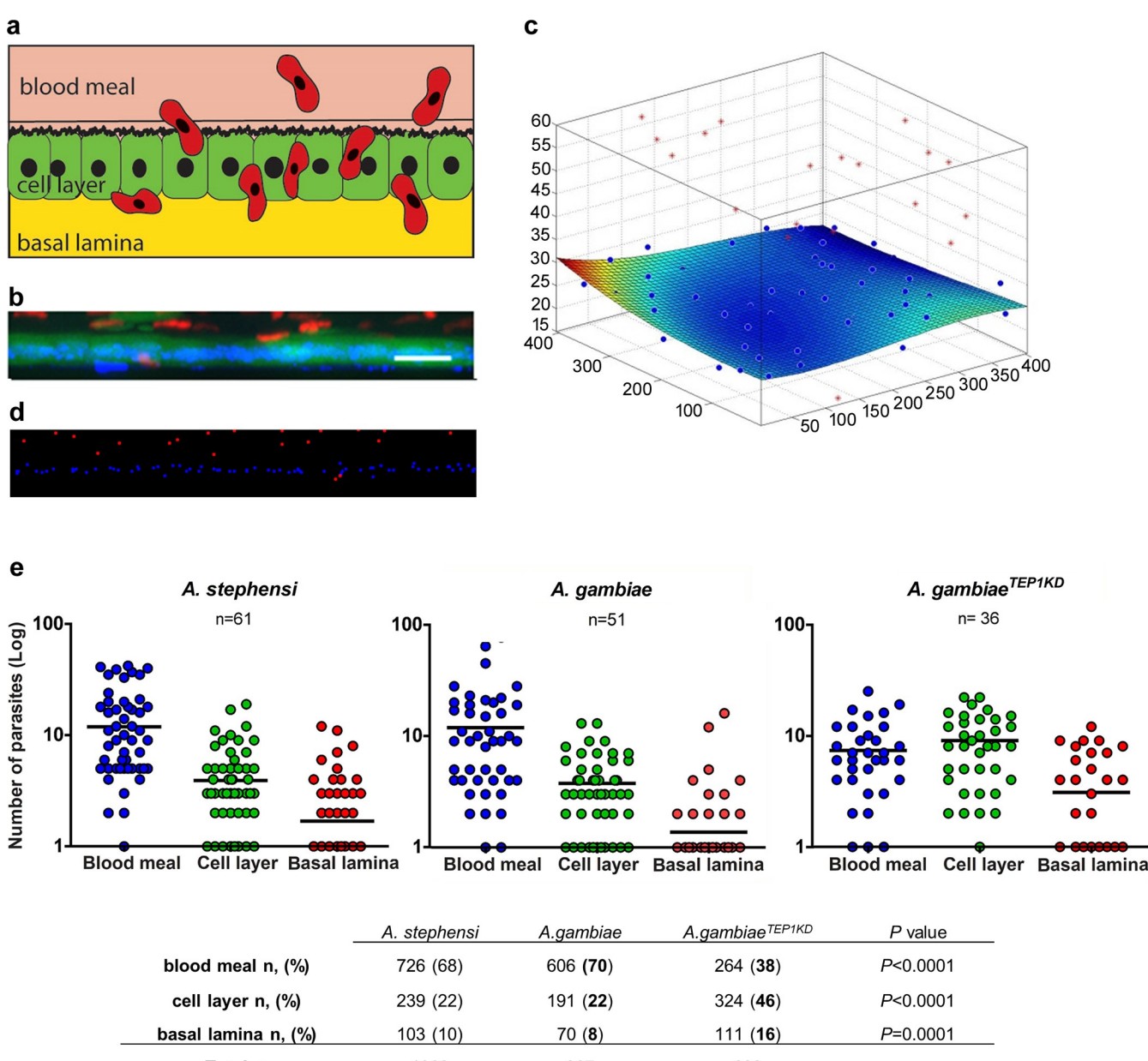

**Fig 2. Positions of the parasites relative to the midgut cells. a**. Schematic representation of the topology in the mosquito midgut. Motile ookinetes (red) traverse the mosquito midgut cells (green) and establish infection on the basal side under the basal lamina. **b**. A representative projection of a cross section of *A. stephensi* midgut, scale bar—50 μm. GFP-positive midgut cells are in green, RFP-positive *P. berghei* parasites are in red, nuclei are labeled by DAPI in blue. **c**. Schematic 3D representation of the same midgut as in (**b**), where the position of the cell layer is calculated relative to the nuclei. Positions of parasites are indicated as red dots, nuclei as blue dots. Deviation of the cell layer from a flat surface is color-coded from blue to red (blue no deviation, red—10 μm). Note the blood meal location of the majority of parasites (above the cell layer). **d**. Representation of nuclei (blue) and parasites (red) in the same midgut as (**b**) after segmentation. **e**. Pooled positions of the parasites from all records at all time points are shown for three layers relative to the midgut cells (blood meal, cell layer or basal lamina) for *A. stephensi*, *A. gambiae* and *A. gambiae* mosquitoes silenced for *TEP1* (*A. gambiae*^TEP1KD). Each dot represents the number of parasites at a given position in a single midgut. The numbers of midguts analyzed (n) are indicated above the graph. Horizontal lines depict the mean number of parasites per position. The table below summarizes parasite distribution inside the mosquito midguts at 18–25 hpi. The percentage of ookinetes in the midguts of *A. stephensi*, *A. gambiae* and *A. gambiae* silenced for *TEP1* (*Ag*^TEP1KD) at each location (blood meal, cell layer, and basal lamina) is shown in parentheses, n is the number of parasites at each position, total n is the total number of analyzed parasites. Statistical analyses were performed by Mann-Whitney t-test and the obtained *P* values are shown.

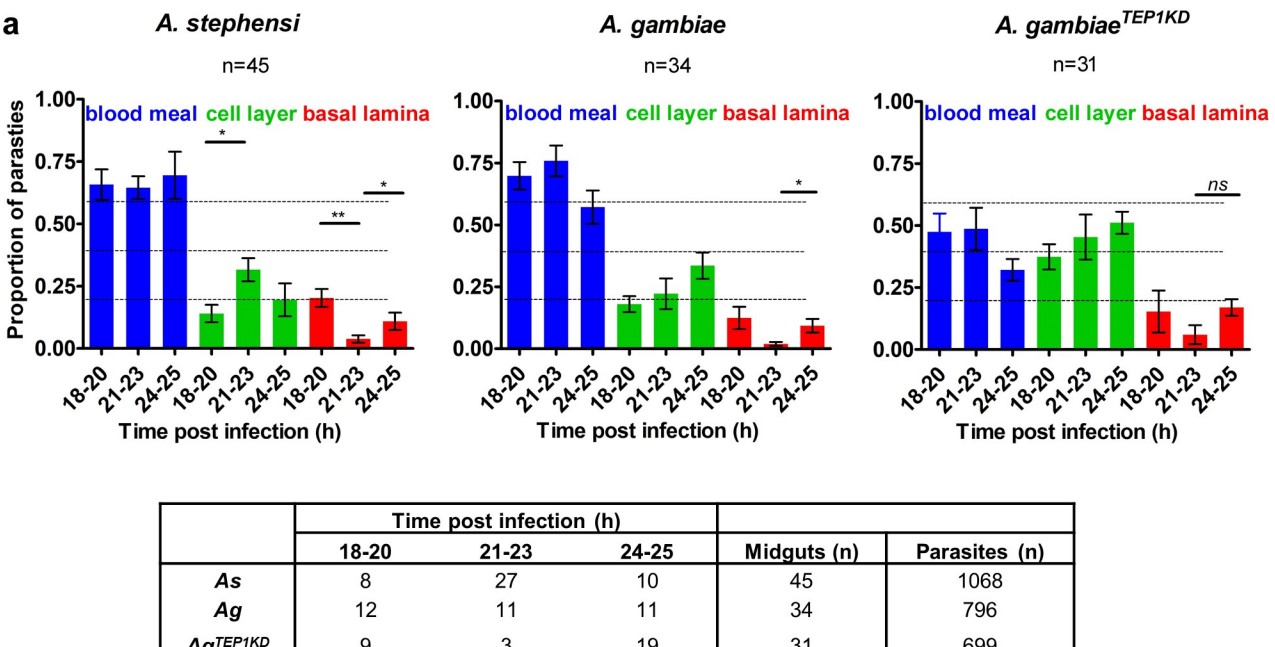

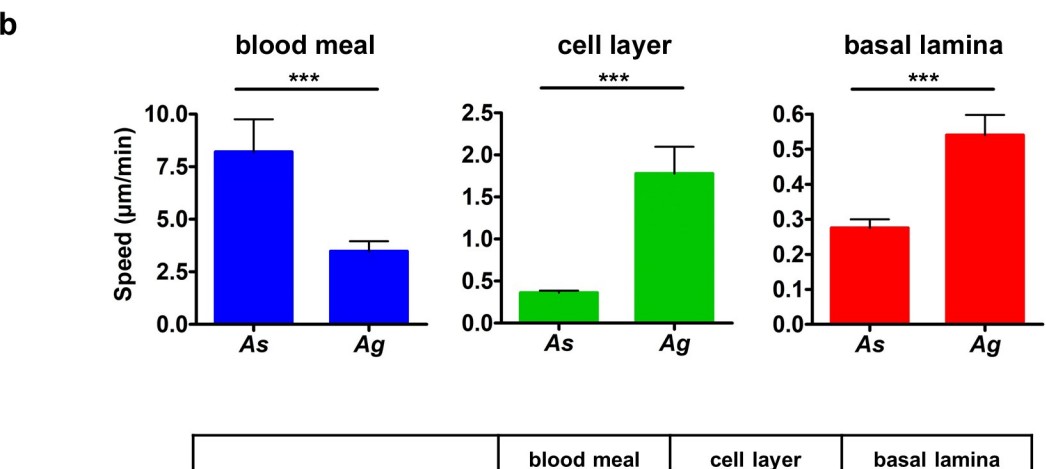

**Fig 3. Kinetics of *P. berghei* invasion of *A. stephensi* and *A. gambiae* midguts. a**. Positions of parasites in *A. stephensi* (*As*), *A. gambiae* (*Ag*) and in *A. gambiae* mosquitoes silenced for *TEP1* (*A. gambiae*^TEP1KD^) between 18 and 25 h post infection (hpi). Plots show the proportion of parasites at each position (blood meal, cell layer, and basal lamina) for three time intervals (18–20, 21–23 and 24–25 hpi). Each bar represents the average proportion of parasites in midguts that contained at least 10 parasites. Parasite positions were calculated by the distance from the cell layer: blood meal for ookinetes detected more than 5 μm above the cell layer; basal lamina for parasites observed more than 5 μm below the cell layer. Statistical analyses were performed by a Mann-Whitney t-test. The table below shows the number of midguts analyzed at each time interval for each mosquito type. **b**. Speed of parasites as function of the parasite position in *As* and *Ag*. Speed (μm/min) was determined by tracking the parasites position over time from the time-lapse series. Four time-lapse experiments were used: guid 1615 and guid 1628 for *As* and guid 1622 and guid 2109 for *Ag*. The table below details the number of frames (n) used for speed calculations. Statistical significance of differences in the average speed at each given position between *As* and *Ag* were examined by the Mann-Whitney t-test and *P* ≤ 0.0001 are designated by three asterisks.

(guid 2115, S1 Table). Some ookinetes were observed within a midgut cell for more than one hour, suggesting that the parasites may remain intracellular for relatively long periods of time without inducing apoptosis. By measuring the parasite speed in the blood meal, cellular layer, and at the basal lamina, we found that the speed of ookinetes carried by the bolus content was the highest as compared to other locations (Fig 3B). Interestingly, the speed of the ookinetes in the blood bolus differed between *As* (8.2 μm/min) and *Ag* (3.4 μm/min) midguts, suggesting some differences in the blood bolus environment. The ookinete spiraling motility in the cellular layer was much slower in both mosquito species, namely 0.36 μm/min in *As* and 1.78 μm/min in *Ag*. The slowest stationary rotation movement of parasites was observed at the basal lamina (in *As*, average speed 0.28 μm/min, guid 2113, S1 Table, in *Ag*, average speed 0.54 μm/min, guid 1622, S2 Table). We noted that the speed of ookinetes within the cellular layer and at the basal lamina was faster in *Ag* than in *As* mosquitoes. This observation indicates important differences in the cellular organization of midguts of the closely related mosquito species.

## Ookinete invasion routes

To characterize ookinete invasion routes, the intra- or extracellular location of the ookinetes at the cellular layer was examined in more detail. We developed an algorithm that classified intracellular, extracellular, and intercellular parasites based on the score of their 3D distance to the four nearest neighboring nuclei of the midgut cells. The score was calculated for each parasite in the cellular layer (Fig 4A and 4B). The parasites with the score between 0–0.45 were defined as extracellular, 0.45–0.55—as intercellular, and higher than 0.55—as intracellular. We noticed a proportion of parasites that was extracellular at all time points in both species (Fig 4C, S5 Fig). When comparing the distribution of intercellular and intracellular parasites, a higher proportion of intercellular ookinetes was observed in *As* (40%) than in *Ag* (20%) (Fig 4D, S6 Fig). These results point to intricate differences in parasite invasion routes between the two species.

## Parasite viability within the midgut

As the transgenic *P. berghei* line used in this study expressed the fluorescence reporter under a constitutive promoter, we were surprised by high variability in the reporter fluorescence levels observed between individual parasites in the same midgut. We examined whether differences in fluorescence intensity correlated with parasite localization and time post infection in two mosquito species. To compare multiple experimental conditions, we normalized fluorescence intensity of each parasite based on the highest and lowest intensity of parasites in each image. For RFP expressing parasites, we found only modest overall differences in mean fluorescence intensities at different positions (basal lamina, cellular layer, blood meal) over time and between the two species (S7–S9 Figs, S8 and S9 Tables). However, we also observed parasites lacking fluorescence that appeared as a black hole on the background of the midgut cells expressing GFP reporter in *Ag* mosquitoes (Fig 5A) that expressed GFP uniformly in all midgut cells. In contrast, irregular pattern of GFP expression in the midgut of *As* [19] made this analysis impossible for this species (S1 Fig). We considered the parasites that lost their fluorescence dying or dead [12,27]. On average, 10–15% of all recognized parasites had no fluorescence and were classified as dead (Fig 5B, S10 Fig). Significant differences in distribution within the cellular layer were observed for live and dead parasites ($P = 0.009$, Mann-Whitney test). While dead parasites were observed at the intercellular and intracellular positions, live parasites were mostly intracellular (compare Fig 5C and Fig 4D, S11 Fig, S12 Fig). This observation is suggestive of a more efficient extracellular killing of ookinetes located within the cellular layer. Interestingly, we hardly detected any dead parasites in $Ag^{TEP1KD}$ mosquitoes, confirming the role of TEP1 in extracellular killing of parasites.

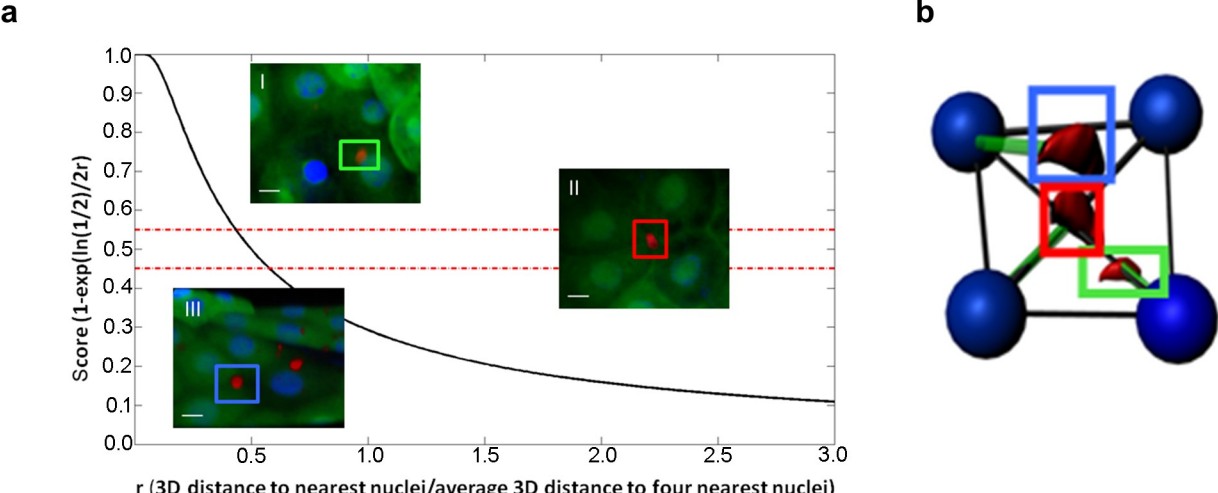

**a** ... **b**

**c**

|  | Intercellular (%) | Intracellular (%) | Extracellular (%) | Parasites (n) |
|---|---|---|---|---|
| *As* (n=61) | 39.0 | 46.7 | 14.2 | 246 |
| *Ag* (n=51) | 27.0 | 56.7 | 16.3 | 178 |
| *Ag*^TEP1KD (n=36) | 28.3 | 56.9 | 14.8 | 332 |

**d**

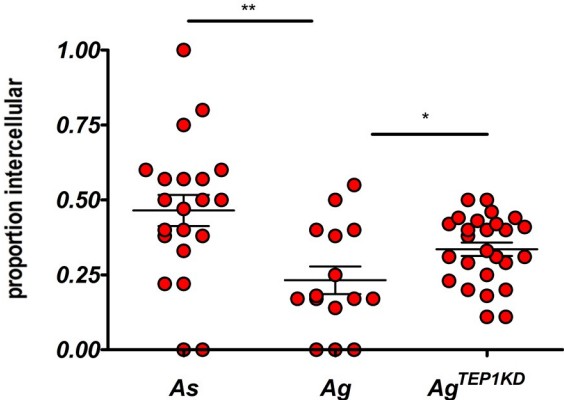

|  | *As* | *Ag* | *Ag*^TEP1KD |
|---|---|---|---|
| *n* | 21 | 15 | 26 |
| Intercellular | 0.46 | 0.23 | 0.33 |
| Intracellular | 0.54 | 0.77 | 0.67 |
| *P* value |  | 0.0033 | 0.03 |

**Fig 4. Parasite distribution in the mosquito midgut.** Parasite positions within the cellular layer calculated relative to the distance of each parasites to the nuclei of surrounding midgut cells. **a**. Calculations of the distance of parasites from the nuclei of the nearest neighboring midgut cell. The score (s) determine whether the parasite is intercellular ($0.45 \leq s \leq 0.55$), extracellular ($s < 0.45$), or intracellular ($s > 0.55$). Example images from a z stack, scale bar = 20 μm: (I) s = 0.74, the parasite (green arrow) is intracellular; (II) s = 0.45 (red arrow) the parasite is intercellular and (III) s = 0.36, the parasite is extracellular (blue arrow). **b**. Schematic representation of parasite (red) and nuclei (blue) positions with distances (green lines) used to calculate distances from the nuclei. **c**. Positions of parasites within the cell layer in *A. stephensi* (*As*), *A. gambiae* (*Ag*) and *A. gambiae* mosquitoes silenced for TEP1 (*A. gambiae*^TEP1KD). The table indicates the percentage of parasites at each position for each mosquito. The number *(n)* indicates the number of midguts analyzed for each mosquito genotype. **d**. Comparison of the proportion of intercellular parasites between *As*, *Ag* and *Ag*^TEP1KD. Each dot represents the proportion of parasites detected between cells in a single midgut. Midguts (n) with at least six parasites within the cellular layer were used for analyses. Statistically significant differences between *As* and *Ag* and between *Ag* and *Ag*^TEP1KD revealed by a non-parametric Mann-Whitney t-test are indicated by asterisks (*—P = 0.03; **—P = 0.003). The table details the mean proportions of parasites in each midgut and for each position for *n* mosquitoes.

**a**

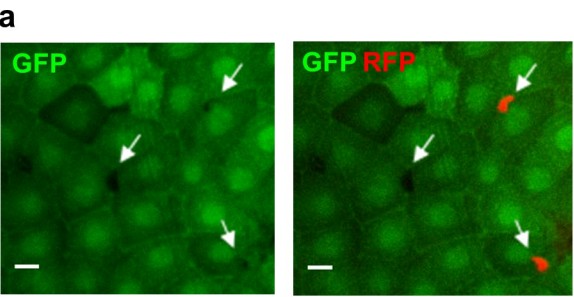

**b**

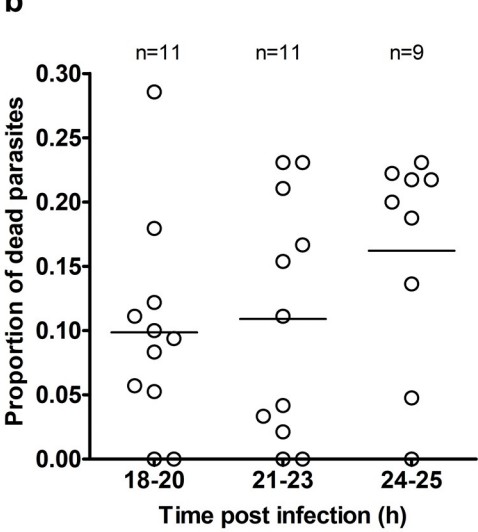

**c**

|  | Intercellular (%) | Intracellular (%) | Extracellular (%) | Total (n) |
|---|---|---|---|---|
| *Ag*<sup>dead</sup> (n=41) | 45.5 | 45.5 | 9.0 | 134 |

**Fig 5. Quantification of dead parasites in *A. gambiae*. a.** Detection of dead parasites within the cellular layer. Due to uniform GFP expression with the midgut cells of the *dmAct5C::dsx-eGFP* line of *A. gambiae*, dead parasites that no longer express RFP could be distinguished in the midgut by their negative signal and a characteristic shape. Shown is a single z-section (scale bar = 20 μm) containing two live RFP-expressing parasites and one dead parasite, indicated by arrows. **b.** The proportion of dead parasites at different time points after *Ag* infection. Midguts (n) that contained at least 10 parasites were used for analyses. Each dot represents a single midgut. **c.** Distribution of dead parasites within the cellular layer. The table shows the percentage of parasites at each position at all time points. All images that contained dead parasites were analyzed. The number (n) is the number of midguts analyzed, total is the number of analyzed parasites.

## Cell damage caused by parasite passage

Midgut regeneration is a natural process of epithelia renovation after a blood feeding, whether infective or not [28]. Blood meal generates a stressful environment as it contains bacteria, reactive oxygen species and digestive enzymes that may damage the midgut cells. It has been previously suggested that invaded midgut cells die after invasion and are expelled into the midgut lumen [29,30] resulting in accumulation of hundreds of extruded cells in highly infected midguts. However, we only once observed GFP-positive midgut cells in the midgut lumen. This result indicates that either dead midgut cells rapidly lose their GFP fluorescence upon

expulsion, or that only few midgut cells are expelled after invasion. To resolve these conjectures, we investigated the integrity of the cell layer using high molecular weight Texas Red-conjugated dextran which is trapped inside damaged cells [31]. In these experiments, the fluorescent dextran was delivered into the midgut by blood feeding mosquitoes on mice injected intravenously with fluorescent dextran several minutes before mosquito feeding. We detected dextran-filled cells (Fig 6A), calculated their position relative to the cellular layer (Fig 6B) and measured the distance to the nearest parasite (Fig 6C). The majority (70%) of dextran-positive cells that contained a parasite in *As* were predominantly detected in the cellular layer. In contrast in *Ag*, dextran-filled cells were observed both in the cellular layer and in the midgut lumen (Fig 6B). As many as 30% of dextran-filled cells in *Ag* were observed in the midgut lumen. Half of these expelled cells contained a parasite (S10 Table). However, in all our experiments, only a single dextran-positive cell was detected in the midgut lumen of *As* mosquitoes shortly after infection. We also observed that at the later time interval (24–25 hpi) in *Ag*, the distance between the dextran-positive cell and the nearest parasite significantly increased as

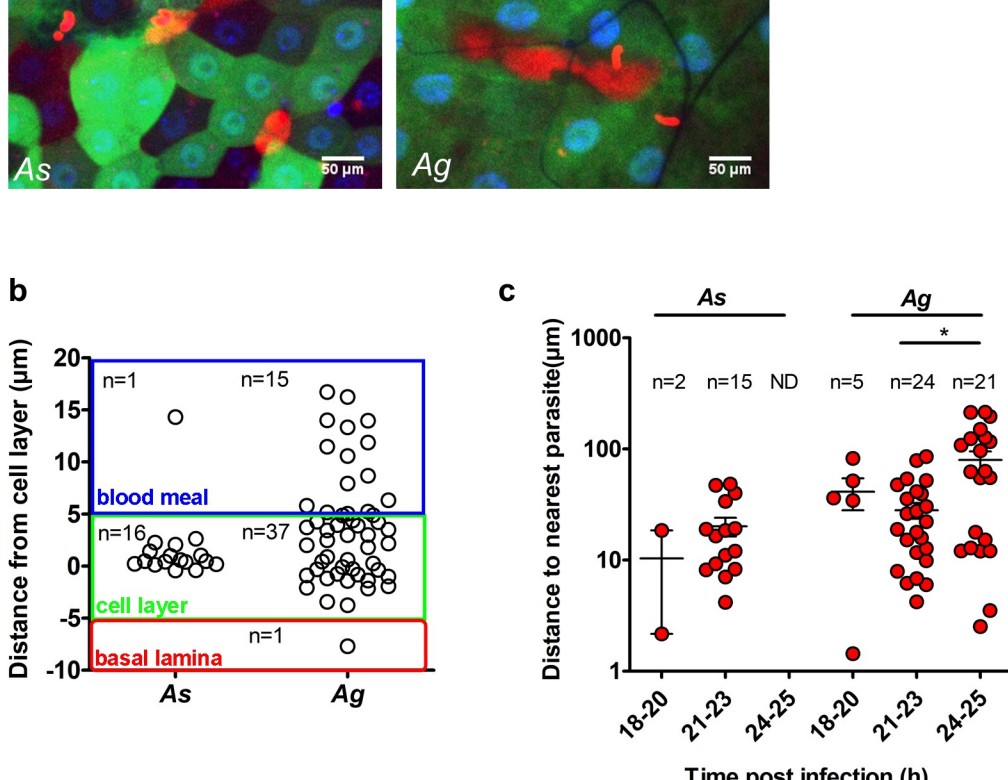

**Fig 6. Quantification of damaged cells. a**. Detection of dextran-positive cells in the midguts of *A. stephensi* (*As*) and *A. gambiae* (*Ag*) mosquitoes. Shown are single z-sections of GFP-expressing dissected midguts. Mosquitoes were fed on mice injected with Texas Red-conjugated dextran. Dextran-positive cells appeared red *(scale bar = 50 μm)*. **b**. Positions of dextran-filled cells in the midgut layers of *As* and *Ag*. Each dot represents a single dextran-positive cell. The graph depicts positions of the dextran-positive cells within the mosquito midgut. Each layer is color coded: blood meal (blue), cell layer (green) and basal lamina (red). The number of dextran-filled cells (n) at each position is indicated. **c**. Distances of dextran-filled cells to the nearest parasite at different time points after infection of *As* and *Ag*. The number of dextran-positive cells analyzed (n) is shown. Statistical analysis was performed by a Mann-Whitney t-test.

compared to the earlier time intervals (Fig 6C). Taken together, these results suggest that damaged cells with or without the parasites are readily extruded into the midgut lumen of *Ag* mosquitoes. It is important to note that while some dextran-positive cells contained a parasite, the majority of invaded midgut cells were dextran-negative, indicating that ookinete invasion damaged and killed only a few midgut cells. The "time bomb" theory of midgut invasion [29] postulates that the parasite passage irreparably damages and kills the invaded midgut cells. This model was further supported by studies of stained sections of *As* midguts infected with *P. falciparum* [30]. The authors reported extensive damage of invaded midgut cells and their expulsion into the lumen. The results of our study suggest that fewer midgut cells are damaged as a result of *Pb* invasion. While 261 parasites were detected in the cellular layer and basal lamina in *Ag* (Fig 2E), we observed only 50 dextran-positive cells (Fig 6C). Similar results were obtained in *As* mosquitoes where for 342 parasites detected in the cellular layer and basal lamina (Fig 2E), only 17 cell were dextran-positive (Fig 6C). We concluded that the majority of midgut cells in *Ag* and *As* survive *Pb* invasion and only few damaged cells are expelled into the midgut. The divergence between this study and the previous reports may relate to the use of different parasite and vector species or may be caused by methodological differences in midgut imaging, namely whole live midguts versus stained midgut sections. Interestingly, in both mosquito species, we never observed more than one parasite in a non-damaged midgut cell, indicating that either parasites refrain from entering already invaded cell, or that entries of multiple parasites swiftly destroy the cells. Similar to the earlier reports, we evidenced chains of connected dextran-positive cells, indicating that ookinetes can traverse multiple neighboring cells before exiting on the basal side of the cellular layer. In conclusion, our results led us to suggest that the route of ookinete invasion for the same parasite is shaped by the mosquito species-specific peculiarities of midgut tissue morphology, physiology, damage and immune responses. Future studies should examine invasion strategies of the human malaria *P. falciparum* parasites in diverse mosquito species.

## Conclusions

By combining live imaging techniques with quantitative bioimage analysis workflow, we uncovered differences in ookinete invasion strategies in two related mosquito species. We found evidence that in both species, the "pioneer" parasites that first reach the basal side of the midgut were rapidly eliminated by the mosquito immune system, and that colonization of the mosquito midgut was initiated at later stages of the infection. High throughput image data analyses of two *Anopheles* species revealed important differences in parasite invasion routes. We showed that the average ookinete speed in the cellular layer is lower in *As* compared to *Ag* mosquitoes. Moreover, *As* midguts contained more intercellular parasites and displayed lower numbers of damaged parasite-harboring cells. These results indicate that faster ookinete speeds and preference for intracellular route may impede parasite survival during invasion in *Ag*, the mosquito species which is more resistant to *P. berghei* infection.

The reported here combination of live imaging and automated image analysis is highly adaptable and can be extended to functional analyses of gene knockdowns, mutations, and drug treatments. Moreover, the image data base and image analysis tools generated by this study offer a powerful tool for studying *Plasmodium* motility in *Anopheles* mosquitoes.

## Materials and methods

### Mosquito rearing

Transgenic *Anopheles stephensi* mosquitoes expressing GFP under the midgut-specific G12 promoter (*pG12::EGFP* [19]) and *Anopheles gambiae* line expressing GFP under the

*Drosophila Acti5c* promoter (*DmActin5c::dsx-eGFP*) [20]) were reared in the lab as previously described [32]. Briefly, mosquitoes were maintained in standard conditions (28˚C, 75–80% humidity, 12/12 h light/dark cycle). Larvae were raised in deionized water and fed finely ground TetraMin fish food. Adults were fed on 10% sucrose *ad libitum* and females were blood-fed on anaesthetized mice. To obtain *Ag* mosquitoes that do not express *TEP1*, the dominant *TEP1* knockdown $Ag^{TEP1KD}$ transgenic line [33] was crossed to *DmActin5c::dsx*-e*GFP* mosquitoes. The $F_1$ progeny had reduced TEP1 expression levels while expressing GFP in the midgut [33].

### *P. berghei* infections

Mosquitoes were blood fed on *P. berghei* infected mice as previously described [34]. *P. berghei* pyrimethamine resistant strain (RMgm 296) constitutively expressed RFP [21]. For the visualization of damaged mosquito cells, mice were injected in the tail vein with 0.1 ml of 5% dextran (3,000 kDa Texas Red-conjugated, Invitrogen) diluted in PBS 10 min prior to blood feeding. Mosquitoes were blood-fed for 20 min on anesthetized mice and dissected between 18–24 h after blood feeding, as indicated in each experiment.

### Confocal microscopy

Immediately prior to visualization, infected mosquitoes were dissected on ice in PBS buffer supplemented with 0.02% DAPI (Thermo Fisher, 4′,6-diamidino-2-phenylindole, 5 mg/mL), and with 0.2% tricaine (Sigma), 0.02% tetramisole (Sigma) to prevent midgut contraction during image acquisition. Blood-filled midguts were placed on 35 mm plastic dishes with glass bottom (Nunc, ThermoFisher). Dishes were mounted on inverted DMI6000 Leica Microscope, equipped with a Nipkow Disk confocal module (Andor Revolution), 20X objective. For time-lapse experiments, samples were visualized for up to two hours at 1 min intervals. The number of 1 μm-stacks, annotated for each image, ranged between 24 and 95 depending on tissue thickness. We noticed that *As* midguts were rigid and sturdy, allowing for longer live imaging. *Ag* midguts were more fragile and tended to move and tear during image acquisition. Live imaging data was collected from *As* (n = 16, S1 Table) and *Ag* midguts (n = 5, S2 Table). We were not able to follow parasites in mosquitoes lacking the immune protein TEP1 due to high fragility of $Ag^{TEP1KD}$ midguts.

### Image analyses

All images were uploaded to a database where they were annotated according to mosquito species and experimental conditions. Images were subjected to bulk analysis as well as manual verification. The annotated image database is accessible to JAVA programming using the Strand Avadis IManage data management software. All data (images and extracted data as text files) are available on cid.curie.fr, Project "Malaria parasite invasion in the mosquito tissues" at https://cid.curie.fr/iManage/standard/login.html. The META data is managed using OpenImadis https://strandls.github.io/openimadis/. Companion scripts are available here: https://github.com/PerrineGilloteaux/MalariaParasiteinMosquito.

The api documentation is accessible under API tab https://cid.curie.fr/iManage/api/client/. The api client jar can be found at https://cid.curie.fr/iManage/standard/downloads.html. For tutorial on access and use of database see: https://youtu.be/mznoB-y99Uo.

Companion scripts include segmentation of parasites, nuclei and quantification of intensities (corrected by background) and were performed using a set of ImageJ Plugins in Java. Analysis of the position of parasites relative to cell layer and statistics were performed with MATLAB.

The data set for a total of 2,557 parasites was collected from 110 independent experiments. More specifically, *As*: n = 45 midguts with 1,068 parasites; *Ag*: n = 34 midguts with 796 parasites; and $Ag^{Tep1KD}$: n = 31 midguts with 693 parasites. There was no bias in the number of parasites per midgut across the time points and mosquito species (S2 Fig). As low infection levels affected quantification of parasite distribution (S3 Fig), only the images that contained at least 10 parasites were included in the analyses.

To determine the position of parasites relative to the epithelial cell layer, the position of the cell layer was modelled for each image as a 3D deformable mesh. The parameters of this mesh were obtained by minimizing the distance between the mesh and the nuclei centroids obtained after 3D segmentation of the nuclei. All data on the fitted meshes against the nuclei position are available in the database as CellLayerFitted_RecordGuid.jpeg from the user attachment panel, where blue spots are nuclei centroids and red stars represent parasite centroids. The positions of nuclei and parasites can be seen by activating the corresponding overlays.

Position of each parasite in Z was corrected according to Z of the nearest point on the mesh (the mesh Z position was created from the polynomial fit for every pixel of the image to increase the resolution of each parasite projection on the mesh). The cell layer was approximated by the average cell diameter, validated by iteration and returned to the data and the classification of parasites. Each parasite had an identifier, and its properties were stored as an attachment in the database so that any computed value can be validated visually. Parasites with corrected positions raging form -5 μm to +5 μm were considered at the cell layer position. Parasites at a distance greater than 5 μm above/below the cell layer were considered in the blood meal or basal lamina, depending on midgut orientation (metadata "reverse" in the database). For each record, a 3D projection in XZ and YZ is available for visualization of the parasite positions (result/bookmarks/xz_yz_visualization).

The position of parasites found within the cell layer was further defined relative to midgut cells. Each parasite was given a score based on a normalized ratio of the 3D Euclidean distance between the center of the parasite and the center of the nearest nuclei, and the average distance of the nuclei center between them (examples in Fig 4A and 4B). The scores were validated by visual examination and are available in the database.

For parasite kinetics tracking at short timescales, movies were first compensated for movement based on the 3D nuclei position. To this end, the 3D movies were cropped around each parasite's X-Y trajectories by projection, and the movement of the cropped image was determined by the 3D drift compensation plugin (Fiji) using the DAPI (nuclei) channel as a reference to estimate the movement. This 3D compensation over time was then applied to the RFP channel showing the parasites. An example of a compensated movie is available at https://youtu.be/690AIgcAIj0. The compensated cropped movies were used to track the parasite over time, after visual validation that the nuclei were correctly stabilized (the original files can be found in the image database: Results/Bookmarks/Compensated movies). All data extracted from the database and used for figures can be found in S1 Data file.

### Ethics statement

The animal work described in this study received agreement #E67-482-2 from the veterinary services of the region Bas-Rhin, France (Direction départementale des services vétérinaires).

## Supporting information

**S1 Fig. *P. berghei* infection intensities in the transgenic *A. stephensi* and *A. gambiae* mosquitoes expressing GFP in the midut cells. a.** GFP fluorescence in the midgut cells of *A. stephensi G12::GFP* line (upper) and *A. gambiae dmActin5c::dsx-eGFP* line (lower) 24 h after

blood feeding. Enlarged are representative 20-fold magnification images showing GFP fluorescence in enterocytes (scale bar—50 μm). **b.** *P. berghei* infection intensities in *A. stephensi* and *A. gambiae*. Oocysts were counted in dissected midguts 7 days post infection. The results of two independent experiments are shown. Prevalence indicates the percentage of infected mosquito midguts in each experiment. Horizontal lines depict median number of oocysts per midgut. Statistical differences between infections of *As* and *Ag* were evaluated by a nonparametric *t* test, *** indicate $P < 0.0005$.
(TIF)

**S2 Fig. Parasite numbers used for analyses in *A. stephensi*, *A. gambiae*, and *A. gambiae* depleted for TEP1 (*A. gambiae*$^{TEP1KD}$).** Each dot represents a single midgut. Similar numbers of parasites were analyzed in all mosquitoes at the indicated time points (h) after infection (hpi), where n is the number of analyzed midguts.
(TIF)

**S3 Fig. Parasite distribution in *A. stephensi*, *A. gambiae*, and *A. gambiae* depleted for TEP1 (*A. gambiae*$^{TEP1KD}$) at different infection levels.** Localization of parasites in the blood meal (blue), cell layer (green) and basal lamina (red) in the midguts grouped by the infection level. Low infection (up to 15 parasites), intermediate (16–35 parasites) and high (more than 35 parasites per image) are compared. Each dot represents the proportion of parasites at a given position in a single midgut. *n* is the number of analyzed images.
(TIF)

**S4 Fig. Temporal dynamics of parasite distribution in *A. stephensi*, *A. gambiae*, and *A. gambiae* depleted for TEP1 (*A. gambiae*$^{TEP1KD}$).** Proportion of parasites found in the blood meal (blue), cellar layer (green) and basal lamina (red) at the indicated time points (h) after infection (hpi). *N* is the number of analyzed images. All analyzed images contained at least ten parasites.
(TIF)

**S5 Fig. Parasite distribution within the cell layer in *A. stephensi*, *A. gambiae*, and *A. gambiae* depleted for TEP1 (*A. gambiae*$^{TEP1KD}$).** Scatter plots depict the score for each parasite at indicated times after infection. Parasites are considered extracellular when the score $s < 0.45$, intercellular for the score $0.45 < s < 0.55$ (red box) and intracellular if the score $s > 0.55$. n is the number of parasites analyzed at each time interval.
(TIF)

**S6 Fig. Parasite positions within the cell layer in *A. stephensi*, *A. gambiae*, and *A. gambiae* depleted for TEP1 (*A. gambiae*$^{TEP1KD}$).** Scatter plots depict the proportion of parasites at each position within the cell layer: extracellular (blue), intercellular (red) and intracellular (green) at different time intervals after infection. Each dot represents a single image which contained at least 6 parasites in the cellular layer. n is the number of analyzed images.
(TIF)

**S7 Fig. Intensity of parasite fluorescence versus position score in *A. stephensi*, *A. gambiae*, and *A. gambiae* depleted for TEP1 (*A. gambiae*$^{TEP1KD}$).** Each dot represents a parasite. n is the number of parasites depicted. No correlation was found between the position of the parasite and the level of fluorescence intensity when intensity is greater than zero.
(TIF)

**S8 Fig. Intensity of parasite fluorescence in *A. stephensi* (*As*), *A. gambiae* (*Ag*), and *A. gambiae* depleted for TEP1 (*Ag*$^{TEP1KD}$).** Bar graphs depict the distribution of parasite fluorescence

intensity at different positions: blood meal (blue), cell layer (green) and basal lamina (red). Parasites from all time points were pooled to calculate the average normalized intensity. Parasite intensity is normalized for each image, intensity ranges between 0.0 and 1.0, where 1.0 is the maximum intensity observed. Statistical significance of differences within each group was tested by one-way ANOVA, and differences supported by $P < 0.0001$ were considered significant.
(TIF)

**S9 Fig. Parasite fluorescence intensity in *A. stephensi* (*As*), *A. gambiae* (*Ag*), and *A. gambiae* depleted for TEP1 (*Ag^TEP1KD*) at different times after infection and at different positions:** blood meal (blue), cell layer (green) and basal lamina (red). Each circle represents a parasite. n is the number of analyzed parasites. Statistical analysis was performed by non-parametrical Mann Whitney test. Only images with more than 10 parasites were analyzed.
(TIF)

**S10 Fig. Quantification of dead parasites in *A. gambiae*.** The proportion of parasites that are considered dead in each image at indicated time intervals after infection. Each dot represents one image, n is the number of analyzed images. All midguts were used for analysis.
(TIF)

**S11 Fig. Distribution of dead parasites within the cell layer in *A. gambiae*.** Scatter plots depict the score for each parasite at indicated times after infection. Parasites are considered extracellular when the score $s < 0.45$, intercellular for the score $0.45 < s < 0.55$ (red box) and intracellular if the score $s > 0.55$. n is the number of parasites analyzed at each time interval.
(TIF)

**S12 Fig. Localization of dead parasites in *A. gambiae* within the cell layer.** Scatter plots depict the proportion of parasites at each position within the cell layer: extracellular (blue), intercellular (red) and intracellular (green) at different time intervals after infection. Each dot represents a single image, n is the number of analyzed images.
(TIF)

**S1 Table. Time-lapse records of ookinete invasion of *A. stephensi* midguts.**
(PDF)

**S2 Table. Time-lapse records of ookinete invasion of *A. gambiae* midguts.**
(PDF)

**S3 Table. Kruskal-Wallis test of differences in ookinete localization between *A. gambiae* (*Ag*) and *A. gambiae* with silenced *TEP1* (*Ag^TEP1KD*) at the indicated time points (h) post infection (hpi).**
(PDF)

**S4 Table. Kruskal-Wallis test of differences in parasite localization in *A. stephensi* (*As*), *A. gambiae* (*Ag*) and *A. gambiae* silenced for *TEP1* (*Ag^TEP1KD*) between the indicated time points (h) after infection (hpi).**
(PDF)

**S5 Table. Kruskal-Wallis test of differences in parasite localization between *A. stephensi* (*As*), *A. gambiae* (*Ag*) and *A. gambiae* silenced for *TEP1* (*Ag^TEP1KD*) at the indicated time points (h) after infection (hpi).**
(PDF)

**S6 Table. Kruskal-Wallis test of differences in parasite fluorescence intensities in *A. ste-phensi* (*As*), *A. gambiae* (*Ag*) and *A. gambiae* silenced for *TEP1* (*Ag^{TEP1KD}*) between the indicated time points (h) after infection (hpi).**
(PDF)

**S7 Table. Kruskal-Wallis test of differences in parasite fluorescence intensities between *A. stephensi* (*As*), *A. gambiae* (*Ag*) and *A. gambiae* with silenced *TEP1* (*Ag^{TEP1KD}*) at all time points.**
(PDF)

**S8 Table. Kruskal-Wallis test of differences in parasite fluorescence intensities between *A. stephensi* (*As*), *A. gambiae* (*Ag*) and *A. gambiae* silenced for *TEP1* (*Ag^{TEP1KD}*) at the indicated time points (h) after infection (hpi).**
(PDF)

**S9 Table. Kruskal-Wallis analyses of parasite fluorescence intensities in *A. stephensi* (*As*), *A. gambiae* (*Ag*) and *A. gambiae* silenced for *TEP1* (*Ag^{TEP1KD}*), in different locations: blood meal (BM), cell layer (CL) and basal lamina (BL) at the indicated time points (h) after infection (hpi).**
(PDF)

**S10 Table. Number of dextran-positive cells in *A. gambiae* and *A. stephensi* mosquitoes at 18–25 h post infection.**
(PDF)

**S11 Table. Summary of phenotypes *A. stephensi* (*As*), *A. gambiae* (*Ag*) and *A. gambiae* depleted for *TEP1* (*Ag^{TEP1KD}*).**
(PDF)

**S1 Data. Summary of all data used to generate the Figure graphs.**
(XLSX)

## Acknowledgments

GV, JŠtáfková, JSoichot and EAL thank M.E. Moritz and C. Kappler for help with the mosquito colony and parasite cultures; and E. Marois for scientific discussions and support. JSalamero and PPG acknowledge the Structure fédérative de recherche santé François-Bonamy and the SERPICO team, are members of the national infrastructure "France BioImaging". Authors thank A. Volohonsky for graphic expertise.

## Author Contributions

**Conceptualization:** Gloria Volohonsky, Elena A. Levashina.

**Data curation:** Gloria Volohonsky, Perrine Paul-Gilloteaux, Jean Salamero, Elena A. Levashina.

**Formal analysis:** Gloria Volohonsky, Perrine Paul-Gilloteaux, Jean Salamero, Elena A. Levashina.

**Funding acquisition:** Jean Salamero, Elena A. Levashina.

**Investigation:** Jitka Štáfková, Elena A. Levashina.

**Methodology:** Gloria Volohonsky, Perrine Paul-Gilloteaux, Jitka Štáfková, Julien Soichot, Jean Salamero.

**Project administration:** Jean Salamero, Elena A. Levashina.

**Resources:** Elena A. Levashina.

**Software:** Perrine Paul-Gilloteaux, Jean Salamero.

**Supervision:** Jean Salamero, Elena A. Levashina.

**Writing – original draft:** Gloria Volohonsky, Elena A. Levashina.

**Writing – review & editing:** Perrine Paul-Gilloteaux, Jean Salamero.

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
