## [Decision Letter · Decision Letter 0]

9 Apr 2020

Dear Prof. Levashina,

Thank you very much for submitting your manuscript "Kinetics of Plasmodium midgut invasion in Anopheles mosquitoes" for consideration at PLOS Pathogens. As with all papers reviewed by the journal, your manuscript was reviewed by members of the editorial board and by several independent reviewers. In light of the reviews (below this email), we would like to invite the resubmission of a significantly-revised version that takes into account the reviewers' comments.

The reviews are consistent in their recognition of the quality of the work. No new wet experiments are requested by reviewers, although you are free to perform them if you feel they are necessary to address reviewer concerns. However, new image, statistical and quantitative analyses are requested, as well as considerable textual changes. All of the major and minor points made by reviewers seem reasonable to us. Several critiques were shared across reviewers, and should be addressed particularly carefully. To summarize briefly, the main recurrent comment areas include:

Situate the work more thoroughly in the previous literature of invasion and motility.Some claims are too strong and/or inadequately justified (e.g. “TEP1 inhibits midgut invasion”, line 112 “new function”, and others), and should be better supported or toned down.Conversely, the apparent contrast of some of the current findings with the previous ookinete time bomb hypothesis, where most invaded cells apoptose and are extruded, was seen to be under-claimed and insufficiently interpreted, and should be treated more thoroughly.There was concern and/or request for more detail about about the imaging analysis ("given the amount of movement in tissue making conclusions from the kinetics difficult"), and related comments on need to strengthen the statistical and quantitative analysis to be more clear and/or convincing.

We cannot make any decision about publication until we have seen the revised manuscript and your response to the reviewers' comments. Your revised manuscript is also likely to be sent to reviewers for further evaluation.

Sincerely,

Kenneth D Vernick

Associate Editor

PLOS Pathogens

David Sacks

Section Editor

PLOS Pathogens

Kasturi Haldar

Editor-in-Chief

PLOS Pathogens

orcid.org/0000-0001-5065-158X

Michael Malim

Editor-in-Chief

PLOS Pathogens

orcid.org/0000-0002-7699-2064

Reviewer's Responses to Questions

**Part I - Summary**

Reviewer #1: The manuscript by Volohonsky et al describes using new advances in microscopy to address the kinetics of Plasmodium ookinete midgut invasion, revisiting questions from ~15 years ago that were never fully addressed. Overall, the manuscript is well-executed and clearly written, while providing new insight into the some of the specific differences in invasion kinetics between different mosquito vectors. However, in its current form, I believe that there are some missed opportunities to improve the introduction and discussion, and would like additional clarification of live imaging methods given the amount of movement over time. I also don’t completely agree with the data interpretation of the TEP1 knockdown and believe that other alternatives should be explored/described in a revised manuscript in addition to other comments.

Reviewer #2: This paper uses imaging tools and an experimental setup involving the use of transgenic mosquito and malaria strains to track, in real time and in vivo, the invasion dynamics of the malaria parasite as it colonises the mosquito midgut, an essential step in completing its life cycle in the mosquito.

These tools will be valuable for other researchers in the field allowing quantitative and qualitative assessment of this process and how it is affected by experimental perturbation.

Given that invasion dynamics vary greatly, not just between parasite:vector combinations, but within vector species, understanding the nature of this process, the pathways involved and their genetic determinants is important.

I therefore think the article has interest for the field, however I think in terms of the conclusions on some of the biology of the process the data is too preliminary and this needs addressing.

Reviewer #3: This manuscript describes the traversal of malaria ookinetes across the midgut of two anopheline species and tests the effect of knocking down Tep1 on this migration. This study is by an accomplished group, the approach to the research question is novel, the amount of work is impressive, and the data significantly increase our understanding of the mechanics of midgut invasion. However, the manuscript is difficult to read, some of the methodologies are difficult to understand, and some of the conclusions appear to be overstated or not supported by rigorous statistics. In general, I have a favorable view of this manuscript, but significant revisions are necessary.

**Part II – Major Issues: Key Experiments Required for Acceptance**

Reviewer #1: - The introduction could be improved to provide more background regarding our current understanding of ookinete invasion, harnessing several papers in the early 2000s that were thoroughly summarized by Baton and Ranford-Cartwright (2005). Although this reference is cited, I think it is a missed opportunity that wasn’t further discussed to illustrate some of the current gaps in knowledge that are addressed in part through the current manuscript. At one point, it was a more prevalent debate as to whether ookinetes took an intra- or inter-cellular route of invasion.

-Similar to the above comment, I think the discussion should also be expanded to place the results of manuscript in greater context of the previous work. Additional references such as Baton and Ranford-Cartwright (2004), Baton and Ranford-Cartwright (2012), and potentially others would improve the discussion. I also think that these data challenge the “time-bomb” model and should be discussed.

-I was pleased to see some of the video links in Table S1 and S2. I didn’t watch all of the links, but am concerned by the about of movement during the time course imaging experiments. In my mind this limits the amount of analysis that can be concluded from these imaging experiments, including any kinetics experiments. How were these data examined in the presence of these movements? How can things like ookinete speed then be determined in these experiments? This seems like a pretty big hurdle in the analysis and interpretation of the data.

-Throughout the manuscript, there are several mentions that TEP1 or mosquito complement impair ookinete invasion. While I do not refute the data provided in several figures of the manuscript, I don’t believe that one can justify that this is entirely direct based on the data provided or other known aspects of previous work from your group, it’s better to leave this open-ended. As a result, I think direct language like “TEP1 inhibits midgut invasion” should be tempered. To me, it seems that these effects are probably a mixture of direct roles of recognition at the basal lamina and indirect roles in the gut given the increased invasion phenotype and TEP1 levels. Loss of TEP1 could significantly influence the composition of the midgut microbiota, where the dysbiosis could potentially help ookinete invasion. I don’t believe that this has been adequately examined. This idea is supported in part as mentioned by Dong et al (2009) for TEP1, and a recent paper by Mitri et al (2020) https://www.frontiersin.org/articles/10.3389/fmicb.2020.00306/full where loss of APL1 influenced the abundance of specific bacterial taxa in the gut. This alternative at minimum should be integrated into the revised manuscript.

-The temporal aspects of ookinete invasion success are interesting. I was intrigued by the discussion of the later timepoint (Lines 177-193) and found this very similar to the temporal aspects of TEP recognition discussed in a recent preprint by Kwon et al (https://www.biorxiv.org/content/10.1101/801480v1) where they examined the role of the BL as a physical barrier for immune recognition. The degradation patterns and repair of the BL are very similar and should be integrated in the manuscript.

Reviewer #2: Specific, substantial areas to address are the following:

Lines 188-196 Speculation on Tep1 depletion and two waves. This is to explain two timepoints but there is nothing to back it up, even Tep1 staining, measurement of Tep levels etc

Line 220 re: differences in speed of motility at lamina and in epithelia in different mosquitoes - why does it point to ‘cellular organisation’ as being the determinant of this. To me this sounds like a difference in cytoskeleton or receptor:ligand combinations is expected. But couldn’t it be some soluble, humeral factor or more general physiological trait? After all, the differences in speed observed in the blood bolus fraction are not expected to be caused by differences in cellular organisation, are they?

Line 229 -232 The parasites with the score between 0 - 0.45 were defined as extracellular, 0.45-0.55 - as intercellular, and higher than 0.55 - as intracellular.

Is this arbitrary - what is the basis for choosing these values? Particularly the intercellular…if, as suggested this is an important category it would be good to have some validation that these are really intercellular

Line 293 “while some dextran filled cells contained a parasite, most midgut cells that we observed to host a parasite were dextran-negative, indicating that ookinete invasion damaged and killed only a small proportion of midgut cells” this, to me, seems to be an important finding but I am surprised it is not given more context. This is not my field but I was under the impression that prevailing wisdom was this ‘time bomb’ theory where nearly all cells through which the parasite passes are extruded and apoptose. How do these results weigh up against that? Is there any way to quantify the number of invaded cells relative to the number of cells that get extruded? i.e. is this parasite-induced extrusion occurring in a minority or the majority of cell invasion events? What is a ‘small proportion’?

Line 310 “ “pioneer” parasites that first reach the basal side of the midgut were rapidly eliminated by the mosquito immune system, and that colonization of the mosquito midgut was initiated at later stages of the infection” again, this would be an important finding but I am not convinced there is enough proof of that in the data reported here in this manuscript.

It seems to me more like a case of back-fitting hypotheses to fit temporal or special observations. That is to say, it is a good place to start investigation of these hypotheses, facilitated by this excellent set of tools, but to accept or reject these would require further experimental corroboration.

Reviewer #3: 1. Lines 138-141: It is unclear how the investigators fitted the cellular layer relative to the nuclei, and how they devised this methodology. This is important because much of the data rely on accurately measuring the location of the midgut lumen, the cell layer, and the basal lamina. More explanation is needed.

2. Lines 143-160, fig 2e: Notably absent here is any type of statistical analysis that validates the conclusions. Were the trials paired (species and KD treatment) so that comparisons can be made? Moreover, the stated conclusion is “an additional role of TEP1 in inhibition of ookinete midgut invasion.” But the number of parasites in the luminal side of A. gambiae and A. gambiae-TEP1KD mosquitoes does not appear markedly different. Instead, the number of parasites in the cell layer and basal lamina are higher in KD mosquitoes. So, this could be interpreted as TEP1 killing the parasites during cellular traversal and not by preventing invasion. This is where a multivariable statistical analysis would pinpoint where the differences are and would allow for more strongly supported conclusions. Lines 261-263, “we hardly detected any dead parasites in AgTEP1KD mosquitoes, suggesting that TEP1 may be involved in killing parasites within the cellular layer”: the author’s words support the statement I made regarding lines 143-160, fig 2e.

3. Lines 184-187, fig 3a: The conclusion is that Tep1 eliminates the decrease in the basally located ookinetes and increases the proportion of parasites within the cellular layer. Again, this is unsupported by statistics. The only statistical test that could be used here is that in the basal lamina there is a significant bimodal effect that is not present (statistically speaking) in KD mosquitoes. But the trend is very much the same, and the reason for non-significance in KD mosquitoes appears to be the higher variance, which means the interpretation could be a type 2 error. In that sense, I do not believe the conclusions that ensue the rest of the paragraph are supported.

4. Section that begins in line 223, fig 4: It is unclear how the investigators arrived at the math used for figure 4 (and figure 4b is difficult to interpret). This is critical because the math is used to determine parasite location. Perhaps adding the axes (luminal, basal, lateral) to figure 4b plus additional text would make this clearer?

**Part III – Minor Issues: Editorial and Data Presentation Modifications**

Reviewer #1: - I am curious with the new microscopy methods and other calculations of ookinete speed provided in the manuscript, can the authors provide any information on how long it takes for ookinetes to traverse the midgut? Even an estimate of minutes vs hours would be informative.

-For the data presented in Figures 4 and 5, it is not entirely clear what is meant by “extracellular”. Is this considered the midgut lumen? Please clarify.

-Data presented in Figures S8 and S9 is more or less glossed over in the manuscript text. It is curious as to why the intensity of parasite fluorescence is lower in intensity in the TEP1 kd. Could this be an effects of “weaker parasites” surviving when they would ordinarily be killed?

-Lines 362-363: Why would TEP1 kd increase midgut fragility?

Reviewer #2: Line 76 needs citations

Line 88 given that the above statement mentions that APL1,LRIM1 and TEP1 all circulate in hemolymph it is not obvious why crossing 'between' epithelial cells to reach basal lamina would mean they 'thus avoid' TEP1

Line 119 state the promoter

Line 146 is there a difference in number of ookinetes between Ag and As though? Apparently not, looking at left and mid panels of Fig 2e. So difference between species is noteable at ookinete to oocyst transition? is this discussed.

Line 152 are these differences significant? P-values and test used?

There appears to be significantly different expression patterns of GFP (Fig S1) between Ag and As, with the latter being very heterogeneous, including large numbers of cells that do not express the marker? Can this confound/bias the results in terms of cells counted/viewed? – I note this is addressed later, somewhat. But could the presence and absence of expression represent different cell types – if so couldn’t this be problematic in drawing general conclusions across the whole midgut epithelium, for As at least?

Line 202 “guid” needs explanation

Line 258 states “Differences in distribution were observed for live and dead parasites within the cellular layer” but the legend for Figure S7 states “No correlation was found between the position of the parasite and the level of fluorescence intensity.” When indeed the figure does seem to back up the main text in that it shows an enrichment of black parasite foci that correlate with the midgut epithelial stack. Connecting this point to the point above about surety of calling ‘intercellular’ parasites, I find it curious that there is not difference between the %dead in the intercellular and intracellular classes, indeed they have identical values. Perhaps some of this is confusion is me getting mixed up with ‘position score’ (relating to inter, intra, extra etc.) and height in stack i.e. basal lamina, epithelial or bolus. Either way, if I’m confused so will some readers be, so I would suggest trying to make this a bit clearer.

Fig 4d is a graph and table of identical I presume. I would leave the graph and label y-axis as “%intercellular parasites”

Line 259 “More dead parasites were found to be located extracellular or intercellular (compare Fig 5c and Fig 4d)” it seems strange to make reader compare values in two different figures to reach a conclusion but if I’ve followed correctly, for all parasites in Ag there is a 27:56.7 ratio of inter:intra whereas for ‘dead only’ it is 1:1 (45.5:45.5). Presumably this is significant?

I am confused about the sample sizes in Fig 5. In panel b there are 31 mosquitoes looked at. Panel c says 41, which I presume refers to the number of mosquitoes. Were they separate experiments?

Line 287 should this be “one dextran filled cell …with a parasite” or is this one of 20 cells found “in” the midgut had dextran? I must say it’s not clear to me how one efficiently recovers or visualises single cells in a cavity as voluminous as the midgut lumen without a large error.

Line 298 “indicating that parasites refrain from entering an invaded cell.” Are there alternative explanations for this observation other than they are prevented from entering - what if two parasites entering the cell ‘burst’ it, or triggered a very rapid apoptosis that meant they were just not observed?

Reviewer #3: 1. Lines 110-112: Was a “new function” of Tep1 really discovered, or did we simply learn more about how Tep1 functions? What is presented seem like a more accurate pinpointing of how Tep1 functions.

2. One of the major conclusions of the study is that “the route of ookinete invasion for the same parasite is species-specific and shaped by…”. This could be true, but wouldn’t it be more likely that the difference in invasion efficiency between the two species leads to the phenotypes observed here? Plasmodium berghei does not share an evolutionary history with either A. gambiae or A. stephensi, so it seems unlikely that different invasion strategies evolved.

3. Is it necessary to abbreviate A. gambiae to Ag, A. stephensi to As and P. berghei to Pb in an online only, open access journal? It makes for a clumsy read.

4. Whenever possible it is best to make definitive statements instead of ambiguous ones. For example, lines 101-103 convey the different rates of development without stating which rate is faster. The answer is known, so why not tell the reader? Another example in the same paragraph: line 108 speaks of “unexpected differences” without providing a clue as to what they are.

5. Line 35: would it be best to use “shaped in part” instead of “shaped”? Undoubtedly there are factors that play a larger role than Tep1.

6. The supplementary figures are presented out of order, making it a more cumbersome read.

PLOS authors have the option to publish the peer review history of their article (what does this mean?). If published, this will include your full peer review and any attached files.

Reviewer #1: No

Reviewer #2: No

Reviewer #3: No
---

## [Editor Report · Decision Letter 1]

23 Jun 2020

Dear Prof. Levashina,

We are pleased to inform you that your manuscript 'Kinetics of Plasmodium midgut invasion in Anopheles mosquitoes' has been provisionally accepted for publication in PLOS Pathogens.

Best regards,

Kenneth D Vernick

Associate Editor

PLOS Pathogens

David Sacks

Section Editor

PLOS Pathogens

Kasturi Haldar

Editor-in-Chief

PLOS Pathogens

orcid.org/0000-0001-5065-158X

Michael Malim

Editor-in-Chief

PLOS Pathogens

orcid.org/0000-0002-7699-2064

Please note that new text added in response to reviewer comments generated a small error that should be corrected. At line 167, it is stated that "Furthermore, depletion of another mosquito complement-like factor APL1 resulted in altered midgut microbiome...". APL1 is a leucine-rich repeat protein, not complement-like.
---

## [Editor Report · Acceptance letter]

7 Aug 2020

Dear Prof. Levashina,

We are delighted to inform you that your manuscript, "Kinetics of Plasmodium midgut invasion in Anopheles mosquitoes," has been formally accepted for publication in PLOS Pathogens.

Best regards,

Kasturi Haldar

Editor-in-Chief

PLOS Pathogens

orcid.org/0000-0001-5065-158X

Michael Malim

Editor-in-Chief

PLOS Pathogens

orcid.org/0000-0002-7699-2064